# The Relationship between Genus/Species Richness and Morphological Diversity among Subfamilies of Jewel Beetles

**DOI:** 10.3390/insects12010024

**Published:** 2021-01-01

**Authors:** Yi-Jie Tong, Hai-Dong Yang, Josh Jenkins Shaw, Xing-Ke Yang, Ming Bai

**Affiliations:** 1Key Laboratory of Zoological Systematics and Evolution, Institute of Zoology, Chinese Academy of Sciences, Box 92, Beichen West Road, Chaoyang District, Beijing 100101, China; tongyijie@ioz.ac.cn (Y.-J.T.); Tsaeyang@126.com (H.-D.Y.); joshjenkinsshaw@ioz.ac.cn (J.J.S.); 2University of Chinese Academy of Sciences, Yuquan Road, Shijingshan, Beijing 100039, China; 3Guangdong Key Laboratory of Animal Conservation and Resource Utilization, Guangdong Public Laboratory of Wild Animal Conservation and Utilization, Institute of Zoology, Guangdong Academy of Sciences, Guangzhou 510260, China

**Keywords:** morphological diversity, species richness, higher taxa, geometric morphometrics, jewel beetles, pronotum, elytron

## Abstract

**Simple Summary:**

Morphological diversity and species richness provide insights into biodiversity and have been studied extensively in recent years. Most researchers have found a positive correlation between these factors in many groups at the local community scale; however, this documented relationship has not always been consistent because of diverse niches and the status of an organism in a given ecosystem. Here we propose a new paradigm for the analysis of higher taxa biodiversity based on a cosmopolitan dataset to further investigate this contradiction. The morphological diversity of 1106 buprestid species from around the world was quantified based on the contours of the pronotum and elytron in dorsal view using a geometric morphometric approach. We found a positive correlation between morphological diversity and genus richness while no significance was found in the species-level test. Furthermore, the correlation between morphological diversity and genus richness is higher than it is in the species-level test. Our results demonstrate the superiority of higher taxa in biodiversity, and that the geometric morphometric approach could quite accurately reveal diversity patterns of the family Buprestidae. These conclusions complement the crucial aspect in several disciplines, including biodiversity, phylogeny and evolutionary strategy.

**Abstract:**

A positive correlation between the species richness and morphological diversity of some organisms has been found in almost all studies at the local community scale. However, this documented relationship has not always been consistent because of diverse niches and the status of an organism in an ecosystem. Global taxon sampling, new morphological approaches, and consideration of more taxonomic categories other than species level are possible methods to further investigate this contradiction. In this study, we proposed a new paradigm for higher taxa biodiversity analysis based on a cosmopolitan dataset. A total of 1106 species from around the world representing all subfamilies and 33% genera of Buprestidae (jewel beetles) were selected to test the correlation between morphological diversity (MD) and genus/species richness (GR/SR) among subfamilies. The MD was quantified by the contours of the pronotum and elytron in dorsal view based on a geometric morphometric approach. The positive correlation between MD and GR was found in all test combinations, but was irrelevant in the species-level test. Interestingly, the correlation between MD and GR was higher than MD and SR in both pronotum and elytron measurements. Additionally, the MD of the pronotum is obviously higher than the MD of the elytron. Our results demonstrate that the geometric morphometric approach could quite accurately reveal diversity patterns of the family Buprestidae. Future studies on different groups, using more characters, more analyses and detailed biological interpretations, are required to fully understand the relationship between MD and SR.

## 1. Introduction

Biodiversity plays a host of important roles in biosphere operations, which is usually quantified via proxy indices (e.g., morphological diversity, genetic diversity, species richness), and can be used to reveal changes in ecosystems. The use of different biodiversity indices highlights the disparate nature of biodiversity research [1,2]. Among these, morphological diversity (MD) and species richness (SR) have often been used to explore patterns of biodiversity in ecosystems [3,4]. MD reflects biological, evolutionary and ecosystem functionality [5] whilst SR reveals the number of species in a community, geographical area or evolutionary unit and it is one of the most widely used indices of biodiversity. According to the information contained in both MD and SR, analysis of the relationship between these two indices has been applied to most animal groups, from unicellular organisms to mammals, as well as many plant groups [6,7,8]. Most studies compared the MD and SR of particular geographic populations [9,10,11,12,13,14,15], and revealed that MD was positively correlated with SR [16,17,18,19]. However, this documented relationship has not always been consistent because of diverse niches and status of an organism in an ecosystem [11,13]. This suggests that the relationship between these two indices is not as simple as initially thought. For instance, research on leaf-litter ants found the opposite trend between MD and SR [20]. There are three main reasons for this contradiction. Firstly, there are many shortcomings for local sampling. A global dataset is a great way to solve the dataset incomparability problem, and at the same time catch a glimpse of the organismic biodiversity by excluding the geographical ecological interference. Few studies have focused on the comparison of these different facets simultaneously at a large continental scale [16,21], while examination based on cosmopolitan datasets are lacking entirely. Secondly, the approaches used to analyze data should be updated based on methodological developments. Previously, only a traditional morphometric approach was used to measure of MD in almost all previous studies, while geometric morphometric, a quantitative and powerful approach [22,23,24], was rarely applied. Lastly, higher taxa should be considered as a possible criterion when studying an organismal system, instead of just the richness at the species level. In this study, we would like to erect a paradigm to infer the relationship between MD and genus/species richness (GR/SR), which could be a turning point to overcome the problems mentioned above.

The beetle family Buprestidae (jewel beetles) was selected as a test group because it is one of the most successful beetle clades, with more than 14,000 described species, and has a fairly complex life history, thus improving their adaptability to the environment [25,26]. Owing to the huge number and diverse array of morphological characters, buprestids have received comprehensive attention from researchers [26,27,28,29,30]. Their global distribution and complex biological niches make the Buprestidae an excellent group for biodiversity research. However, up to now, there has been no investigation focusing on the relationship between morphological diversity and species/genus richness within higher taxa of Buprestidae based on a large dataset containing information from around the world.

The morphological diversity was quantified by the shape in dorsal view of the selected species, represented by the outline of the pronotum and elytra, based on a geometric morphometric approach. Different from the previous studies which focused on the mandibles, antenna, genitalia, hind wing, etc., that strongly reflect the selective pressure of the environment (e.g., food, flight behavior, sexual selection, etc.) [23,31,32,33,34], the pronotum and elytron were selected in this study because of their relatively neutral reflection of the environment. The richness of the species and genus levels were applied as the criteria, and comparisons of richness among subfamilies of Buprestidae were conducted in this study.

The aim of this study was to explore whether the correlation between morphological diversity and genus/species richness in higher taxa was consistent with species-level studies of local communities. Test indices were explored based on a large dataset covering 1106 species of Buprestidae, including all 6 extant subfamilies and 173 genera, representing 8% of all known species. The aim of this paper was not only to reveal the expression of biodiversity according to all the subfamilies of Buprestidae around the world as a whole, but also to pave a new way for research on other groups and the relationship between MD and SR.

## 2. Materials and Methods

### 2.1. Taxa Examined

This study analyzed 1106 species (1215 specimens) from all six extant subfamilies (Polycestinae, Chrysochroinae, Buprestinae, Agrilinae, Julodinae and Galbellinae) [35] covering 173 buprestid genera (approximately 33% of all described extant genera) from around the world (Table 1).

### 2.2. Genus/Species Richness

The genus/species richness value was obtained by counting the number of genera and species in different test subfamilies (Table 1). The objectivity of sampling standard was given priority in this study. The described genus/species richness of each buprestid subfamily around the world was obtained [28,36,37,38,39,40,41], then the proportions obtained by comparing the genus/species richness to the total genus/species in each test and described subfamily were computed using SPSS Statistics (Version: 26) to visualize the proportion of species and genera sampled in our dataset versus the number of species and genera described (Figure 1) [42].

### 2.3. Morphological Diversity

The morphological diversity was quantified based on the shape of the pronotum and elytra, which represent two major parts of the body structure in dorsal view and have typically been used as an index for body size, and both of which contain evolutionary information [43]. The head, which is often not visible in dorsal view in buprestids, was excluded in this study. Furthermore, the pronotum was selected because it supports the prothorax’s muscle system and the locomotion of the prothoracic legs [44], whilst also affording protection for the pterothoracic segments as the attachment of muscles [45] coordinates the movement and stretching of the neck, e.g., click beetles (Elateridae) and stag beetles (Lucanidae) [22,46]. The forewings of most beetles were chitinized into elytra during their long evolutionary history. These hardened structures cover the dorsal surface of the beetles thus protecting the membranous hindwings and their ability to fly [28]. In addition, different microstructures both on the surface and inside of the elytra also account for water retention and respiration [47].

The studied images were collected from the collections of the IZAS (Institute of Zoology, Chinese Academy of Sciences), NMPC (National Museum, Prague, Czech Republic), MNHN (Museum National d’Histoire Naturelle, Paris, France), NHML (The Natural History Museum, London, UK), USNM (United States National Museum of Natural History, Smithsonian Institution, Washington, DC, USA), and sources in the literature [25,48,49,50]. Standard dorsal images were selected for this study. To facilitate accurate representation, images were only used if the pronotum and elytron were not covered or blurry and the images possessed adequate resolution (the smallest one was 90 pixels) (Appendix A) [51].

Two curves were extracted from the left contours of the pronotum and elytron to represent their external forms (Figure 2). Curve One was collected from the middle of the anterior margin of the pronotum and ended up at the middle of the posterior margin of the pronotum. Curve Two started from the anterior margin of the left elytron. Each curve was resampled into 25/50 equally spaced semi-landmarks, respectively (Figure 2). All curves and semi-landmarks were digitized with tps-Dig 2.05 [52]. The format of data files used for morphological analysis was achieved by converting semi-landmarks into landmarks [53] in text files for the subsequent analysis: the curve number and point number for each sample were deleted, then landmark numbers were replaced by point numbers [22,23,25]. Principal component analysis (PCA) and the geometric modelling in mathematical spaces formed by the PC axes were used to interpret the outline shape variation of the pronotum and elytron in advance [54,55]; the shape-deformation patterns (expressed by the first three principal components, PC1–PC3), which indicated the morphological variation trend corresponding to the test groups scattered in the morph-space (PC coordinates’ space), were calculated at coordinate positions spaced at equal intervals along each PC axis from the minimum to maximum values of the projected shapes (Figure 3). All the patterns were set to change from red to blue hue on the positive direction of each principal component axis, making the deformation of the test traits displayed on the axis clearer. The morphological diversity was quantified as the total Procrustes variance using MORPHO J 1.06a (Table 2, Figure 4) [22,56]. The Procrustes variance measures the dispersion of all observations around the mean shape of the respective taxa. The exploration of whether the correlations between morphological diversity and species richness was positive in higher taxa was achieved through the Spearman correlation coefficient (single tail) in SPSS Statistics (Version: 26) (Table 3, Figure 5) [42]. In addition, the correlation analysis of the morphological diversity of the pronotum and elytron was accomplished in SPSS Statistics (Version: 26) (Figure 5) [42].

## 3. Results

### 3.1. Morphological Variation of the Pronotum and Elytron

The first three principal components (PCs) accounted for 88.109% of observed shape variation for the pronotum and 91.348% of observed shape variation of the elytra (Figure 3). Along the positive direction of the first PC axis, the left contour of the pronotum became stretched longitudinally and diminished horizontally with the posterior angle shrinking inwardly horizontally and the anterior angle extending outwards, and the entire pronotum becoming markedly more quadrate. Along the positive direction of the second PC axis, the pronotum shape changed from an inverted trapezoidal to trapezoidal outline, with the produced posterior angle and retracting posterior margin making it sharper, while the anterior angle was diminished and became less distinct. Along the third PC axis, the entire half pronotum contour tended towards a triangle with the anterior angle diminishing and becoming less distinct and the posterior angle extending outward horizontally.

Along the positive direction of its first PC axis, the half contour of the test elytron became more prominent, the posterior margin protruded, and the anterior margin stretched horizontally. The shrinking smaller scutellum causes the anterior margin to become longer while the base of the lateral margin becomes protruding outward originally and the rear part becomes contracted inward. Along the positive direction of the second elytron PC axis, the overall morphology of the elytron became narrow and long. This pattern was contrary to the previous one: the posterior border of the elytron expanded laterally, while the anterior angle broadened outward. The expansion of the scutellum edge led to a more inward trend of the anterior margin of the elytron. Along the positive direction of the third elytron PC axis, the smaller scutellum flattened the front margin and, at the same time, the posterior border of the elytron contracted inward, leading the overall morphology of the elytron to become an inverted triangle.

### 3.2. Morphological Diversity among Groups

The Agrilinae exhibited both the highest pronotum shape diversity (0.0187) and elytron shape diversity (0.0062), with the highest genus richness in our study (Table 2, see also Figure 4). Similar elytron diversity was found in the tests of Buprestinae (0.0026) and Chrysochroinae (0.0026), which indicate similar test genus richness. The Galbellinae, a small group containing 84 described species in the world fauna was found to have the lowest pronotum morphological diversity (0.0046) and fairly low elytron diversity (0.0011). However, the relationship between morphological diversity and species richness was not always consistent. Polycestinae was found to have a very high pronotum and elytron diversity (0.0150/0.0057) with a comparatively small number of test species (120) and only 26 test genera. Its elytron diversity (0.0057) was much higher than it in the test of Buprestinae (0.0025). Similar to the test of Chrysochroinae, the Julodinae was found to have a high pronotum diversity while its test genus and species richness were the lowest.

In addition, morphological diversity was also found to vary between the pronotum and elytron of the same taxon. The changing trends of characters and species richness values were not always highly associated: elytron diversity of the Julodinae was relatively high compared with Galbellinae despite showing the lowest test elytron diversity.

### 3.3. Correlation Analysis of Test Subfamilies Under Buprestidae

The Spearman correlation coefficient was applied to the test parameters, which revealed the correlation between morphological diversity and genus/species richness, and is shown in Table 3. Three aspects of biodiversity were revealed through correlation-tests in this study. Firstly, the correlation between morphological diversity and genus richness among test subfamilies was found to always be consistent (*p* < 0.05 in pronotum-genus test and *p* is close to 0.05 in elytron-genus test, respectively), while in the species-level test of the pronotum and the elytron, the correlation test result was non-significant (*p* > 0.05). Secondly, the correlation between morphological diversity and species richness in the pronotum test was found to be weaker than it in the elytron test. The coefficient value was 0.714 in pronotum (genus-level) test and became 0.771 in elytron (genus-level) test. It was 0.371 in the pronotum (species-level) test and 0.543 in the elytron (species-level) test. Thirdly, the degree of correlation between parameters was found to be superior in the higher taxonomic category (Spearman correlation coefficient *r* = 0.714/0.771 in the genus-level test; *r* = 0.371/0.543 in the species-level test). In addition, four best-fit lines (including the 70% equal frequency region) used through ordinary least squares (OLS) regression analysis were processed to macroscopically visualize the correlation coefficient situation based on the regular aggregation degree of the test groups (Figure 5). The distribution of test groups was gathered on both sides of the best-fit line in the genus-level test. On the contrary, the distribution of test groups was more random and scattered in vector-space in the species-level test.

The correlation between the morphological diversity of the pronotum and elytron was analyzed (Figure 5). In the space constructed by the morphological diversity of the pronotum and elytron, all test subfamilies were distributed on both sides of the best-fit line.

## 4. Discussion

### 4.1. Morphological Diversity and Species/Genus Richness of Buprestidae

This study has demonstrated the positive correlation between MD and GR, based on the global sampling of the beetle family Buprestidae, for the first time. Our results confirmed the previous conclusion that the morphological diversity was strongly related to genus category [57]. Based on our dataset, for the species level, no correlation was found between MD and SR. Additionally, we found the morphological diversity of pronotum in each test group was higher than the morphological diversity of the elytron. It is difficult to make a definitive conclusion about why these two characters are not consistent and further studies are needed to investigate it properly. Based on the current knowledge and the results of this study, it is likely explained by the different selective pressures on the pronotum or elytron. Biological diversity can be quantified in terms of species richness and morphological diversity [58], both of which are dependent on ecological variability and species interactions [59]. As a result of selection-based feedback due to the variability of ecological factors, new forms emerge as a result of biological evolution [60]. In buprestids, the posterior margin of the pronotum is adapted to fix the elytral base in repose and in flight and constitutes a rather complicated locking mechanism. This has been proven by the morphological variation of test traits, and the linear correlation between morphological diversity of the pronotum and elytron in this study. Furthermore, since the pronotum is not fully restricted by flight machinery, its higher variation is also influenced by buprestid feeding behavior and the movement mode of the head due to the association of dorsally originating cervical muscle and cervical sclerite muscle between the head and prothorax. Usually jewel beetles feed on diverse plant parts; for instance, the adult and larval leaf-miner buprestids feed on the foliage of the host plant and/or visit flowers to feed on pollen and nectar in the case of adults, and there is a unique feeding strategy employed by *Xyroscelis crocata* (Polycestinae: Xyroscelidini) [61] whereby the adults feed on the sap of the host plant *Macrozamia communis* and the larvae feed within the frond-stalks of cycads [62]. The functional diversity of the pronotum increases under the influence of the feeding process and the flight process, while the external morphology is highly divergent.

Different morphologies are associated with diverse functional aspects of niches [63,64] and both the evolution and development of taxa, and these also account for the variable morphological characters of jewel beetles [40]. In this study, complex and irregular relationships were found among different groups in the subfamily categories along with their diverse habits and quite different coevolutionary modes. The subfamily Agrilinae comprises nearly a half of all known Buprestidae species and is composed of four tribes and 23 subtribes [26,65], with a very high support rate that separates it from other subfamilies based on previous phylogenetic trees [35]. Except for the large genus *Agrilus* with more than 3000 species [66], which is distributed in all geographic regions, few genera are distributed in more than two geographic regions, such as *Aphanisticus*, *Sambus*, *Habroloma* and *Trachys*. Many groups are only distributed in a single region. This may indicate that most genera of this subfamily have evolved relatively special morphologies to adapt to different plant communities in different biogeographic regions, occupying different niches. At the same time, the larvae of this subfamily exhibit three major feeding habits: xylophagous type, stem-miners and leaf-miners [26,65,67]. The different habits of the larvae presumably affect the external morphology of the adults to some extent; for instance, the Tracheini larvae are leaf miners and the adults of this tribe are wide and short. Most adults are narrow with stem-miner larvae in the Aphanisticini. The diverse feeding habits may also play an important role in driving the diverse morphology of Agrilinae [68,69]. Species in Julodinae are nearly cylindrical, tapering towards the posterior end. Only 141 species have been recorded around the world, distributed in the Afrotropic Region, Palaearctic Region and Oriental Region [26]. Moreover, more than half of the species have a scattered distribution with a large span in geographical latitude (e.g., *Julodis algirica*, *Julodis angolensis*, *Neojulodis bequaerti*). The wide distribution and different geographical environments have presumably led to the complex morphological structure of this group, which is reflected in the morphological diversity test of the pronotum. Galbellinae has only one genus with 84 species from around the world, which is mainly distributed in the Afrotropical region, resulting in the low morphological diversity of the elytron. The larvae of Buprestinae bore into weak wood or dead plant parts, and the adults feed mainly on flowers or leaves. Members of this subfamily are also distributed in the major seven geographic regions, but, with the exception of a few taxa (e.g., the genera *Anthaxia*, *Melanophila*, *Chrysobothris*), most genera are limited to one geographic region [26,65], which may explain the high species richness and pronotum morphological diversity in this family. Chrysochroinae are widely distributed in all the geographic regions; most species are concentrated in the Afrotropical region, Oriental region and Neotropical region. The beetles in this subfamily are mostly large and often covered with a secreted waxy substance. Similar to the Buprestinae, it has a relatively high morphological diversity, perhaps corresponding to the wide distribution. Although the Polycestinae is relatively small in comparison to other subfamilies such as Buprestinae and Chrysochroinae, the members of this taxon are also widely distributed in the seven major geographic regions. Beetles in Polycestinae feed on different plant parts; some genera like to visit flowers, such as *Acmaeoderella* and *Acmaeodera*, whilst others feed on leaves. Furthermore, significant morphological changes of body shape were also found in some taxa (such as *Paratrachys*) [27,28,49]. In this study, we found that the test traits of Polycestinae had a diverse morphological variation, which was inseparable based on biological characteristics.

### 4.2. Geometric Morphometrics in the Biodiversity Measurement of Higher Taxa

By using geometric morphometrics, we have here found that a large amount of deformation occurs on specific parts of the pronotum and elytra. In particular, the posterior angle of the pronotum and the scutellum edge attached to the anterior ledge of elytron, which showed the relevance of the morphological function of these above-mentioned traits in morph-space. Excluding the influence of size and orientation factors through Procrustes fit, we found that the test traits cover a large number of variable factors in morphology whilst the morphological diversity obtained based on these deformation factors was also closely related to genus richness. Therefore, these morphological measurements of shape have the potential to become an important part of morphological diversity indices. Nevertheless, for most statistical morphometric measurement-design strategies, the acquisition of two-dimensional linear distances from extremal points of specimens is still the main method [70,71]. Biostatistics methods began to be widely used in morphological data analysis in the middle of the 20th century and eventually evolved into the widely used method of morphometrics. This method mainly compares values such as linear measurement distance, including angle, area, weight, and ratio between variables, and sometimes more discretized results were obtained through logarithmic transformation or coordinate transformation [53,72]. Most biodiversity studies involving morphological diversity are based on these kinds of extraction and measurement methods in the sampling of traits. For instance, Triantis selected the range of shell height and width as morphological indices for analyzing the correlation between biodiversity and biogeographic/climatic factors [57]. Pigot [73] combined data on morphological and ecological traits for 523 species of passerine birds distributed across a single elevation transect in the Andes. They analyzed seven traits to represent the morphological diversity of passerine birds, e.g., beak length, width and depth, tarsus length, wing length and tail length. Moreover, Mindel [17] also determined the functional richness and functional divergence of fish assemblages along the continental slope by using eight traits which included the total length, head length, tail height and eye position.

It is true that linear distances play a certain role in reflecting the growth rate and evolution of organisms, whilst it is scalar magnitudes that can be used to represent size, but contain no shape information whatsoever [53]. In addition, traditional morphometrics has many problems, such as subjective factor interference and the fact that size and shape cannot be analyzed separately [74]. Excluding the shape information, statistically measured data cannot fully represent the morphological diversity of geographic populations, and the complex and changeable ecological factors in geographic regions and different artificial sampling methods may also have led to the unpredictable results of many studies on the biodiversity of local faunas [11,75]. As a more commonly used quantitative analysis method, geometric morphometrics calculates the geometry between the representative points. It can better describe the shape and position of morphological characters based on landmark analysis [24,76]. The deformation amplitude and the phenetic similarity between test samples can be indicated through the Procrustes distances [53,77], in which the variation of sampled shapes can be separated and combined with principal component analysis. The geometric morphological technique has been gradually used in biodiversity: Farre [24] selected 27 landmarks (and also included semilandmarks) based on anatomical, ecological and taxonomical indices to evaluate the usefulness of the geometric morphological indices in studies of fish assemblages by analyzing their correlation between biodiversity indices; Neige [78] pointed out 15 landmarks on the different parts of the cuttlebones for constructing morphospaces and then compared these indices with species richness. With the help of geometric morphometrics, we obtained the deformation status of continuous traits whilst confirming the feasibility of geometric spatial information of the traits as a morphological diversity parameter. In this way, the trait association phenomenon found in traditional taxonomy was visually displayed in the form of geometry and mathematics.

Some scholars believe that higher taxonomic levels are questionable and contain certain subjective elements, and the inconsistent level boundaries make the taxa defined in this framework lack actual evolutionary significance [79,80]. However, Williams [81] found the superiority of the higher category, and therefore suggested its suitability for diversity research [82]: the lower the quantity, the more uneven the distribution of species in each unit [83,84,85], and the time efficiency in dealing with those hyper-diverse, yet small-sized, taxa. As early as 1953, Simpson [86] used the genus level as an example to prove that the higher-level elements do effectively reflect the natural evolutionary process, and experiments in recent years have also confirmed this [87,88]. Taking Buprestidae as an example, based on the morphological data from 1106 species of six existing buprestid subfamilies across the globe, we found that the degree of correlation between indices in the genus-level test was higher than it in the species-level test. At the same time, we found that there was a correlation between the morphological diversity of subfamilies and genus richness but no significance in the species-level test, thus illustrating the higher taxonomic category was the more important variable determining morphological diversity [82]. This situation was also confirmed in other experiments. For instance, Triantis [57] found that the test results of genus-level elements were better than species-level elements by studying land snails. Under this “Top–Down” taxonomic rank, the higher-category has a higher level of dominance than the lower-category. Moreover, morphological delineation analysis based on high-level taxonomic ranks has been found to be consistent with the results of molecular phylogenies in the study of mammals. For instance, Jablonski [89] demonstrated that morphologically defined genera showed a strong correlation of body size and latitudinal range with genera defined on the basis of phylogeny. This implies that, although the higher category has certain limitations, their use in large-scale analyses of morphology and biodiversity is unlikely to be misleading.

In this study, we report a positive correlation between MD and GR based on similar sampling proportions in based on test jewel beetles. However, our results also highlighted the unconformity between species richness and the change trend of morphological diversity in certain groups (e.g., Chrysochroinae), which suggests that the inconsistent correlation between MD and GR/SR could be caused by the influence of other categories, such as tribe rank.

### 4.3. Biodiversity Measurement Based on Cosmopolitan Dataset

Due to the variable ecological niches influences by the latitudinal and elevational gradients throughout world ecosystems, the species richness of organisms decreases from the equator to the north and south poles [90,91]. At the same time, the fauna of an environmental area has also produced a certain spatial structure along with the distribution pattern, which is affected by the environmental gradient [92]. A large number of studies have focused on the spatial distribution and gradient changes of geographic population diversity [6,73,93]. However, the spatially limited distribution of faunas and the similarity of habitat may result in spatial autocorrelation [94,95], which obscures the real situation. Moreover, biological interactions and random processes of local populations make ecological research more complicated [96]. The environmental factors of ecological importance that shape biodiversity also differ across scales [97,98]. The ongoing debate between environmental gradients and the factors that determine species diversity and the autocorrelation of spatial structures of species distribution has promoted the research of large-scale biodiversity to a certain extent [94,95].

It is widely accepted that biodiversity research based on large-scale patterns will be influenced by environmental factors (e.g., climate) [99,100,101]; however, this approach shows its strength by avoiding the influence of species interactions and local complex factors [102,103]. In this study, under the condition of ensuring a balanced proportion of test groups, a large dataset and random sampling were used to dilute the influence of regional environmental factors on the shape and richness of the samples. We obtained results similar to the species-level metabiological diversity test of most taxa, by finding a congruence between the morphological diversity and genus richness. Some research has also expanded samples to the entire continent and showed the uniqueness of large-scale biodiversity research methods. For instance, Kuczynski [16] determined the key drivers shaping freshwater fish biodiversity patterns across Europe from 290 European river catchments and found that the diversity indices were not always strongly consistent. By reporting a strong spatial congruence between biodiversity facets for European ants based on 349 communities across western and central Europe, Arnan [96] found that differences in ecological niches caused changes in sample traits which affected the experimental results, whilst spatial factors might also have had an impact on the diversity patterns. Although it is beyond the scope of this study to compare the global scale and local scale between MD and SR/GR, it would be interesting to examine these relationships in further studies.

## 5. Conclusions

In this study, we showed that there was a positive correlation between morphological diversity and genus richness by regarding the Buprestidae (jewel beetles) as a whole test group. We also found that the geometric morphological method could extract the deformations in the morph-space to improve the morphological diversity data, rather than just relying on the traditional morphometric measurements of features. Furthermore, we found that the pronotum morphological diversity was higher than it was in the elytron and correlations between the genus richness and morphological diversity were stronger than the species level tests.

## Figures and Tables

**Figure 1 insects-12-00024-f001:**
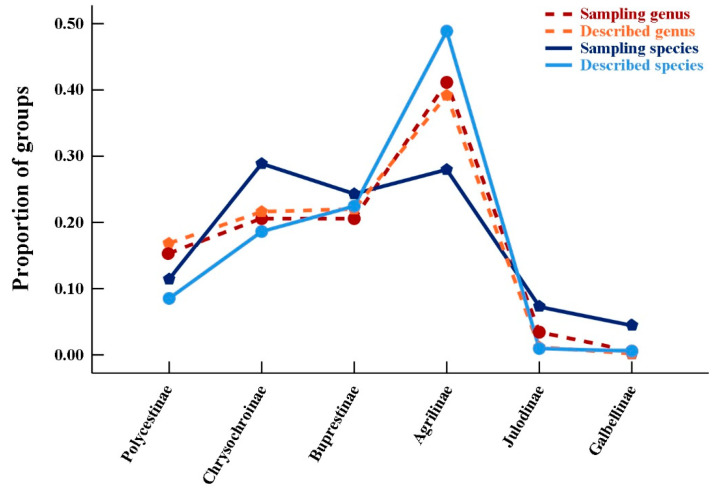
The proportion of genera/species under each subfamily in Buprestidae. Each proportion calculated from the sampling dataset and described dataset is showed, respectively.

**Figure 2 insects-12-00024-f002:**
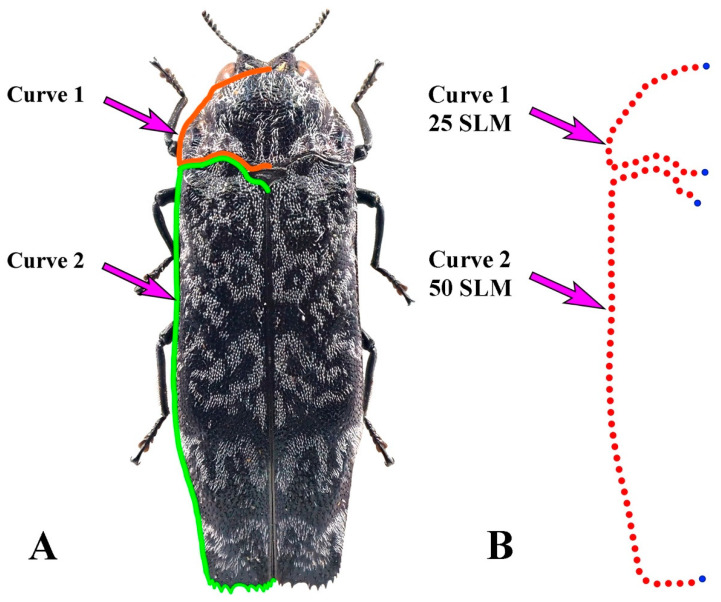
Description of the curves used in the geometric morphometric analysis. The positions selected for the pronotum and elytron curves are represented by *Coraebus spathatus* in dorsal view. (**A**) Curve 1 and Curve 2 are shown in orange and green outlines for the counters of pronotum and elytron, respectively. (**B**) Curve 1 was resampled in 25 semi-landmarks; Curve 2 was resampled in 50 semi-landmarks. The semi-landmarks are shown in red points and the terminal ones are shown in blue.

**Figure 3 insects-12-00024-f003:**
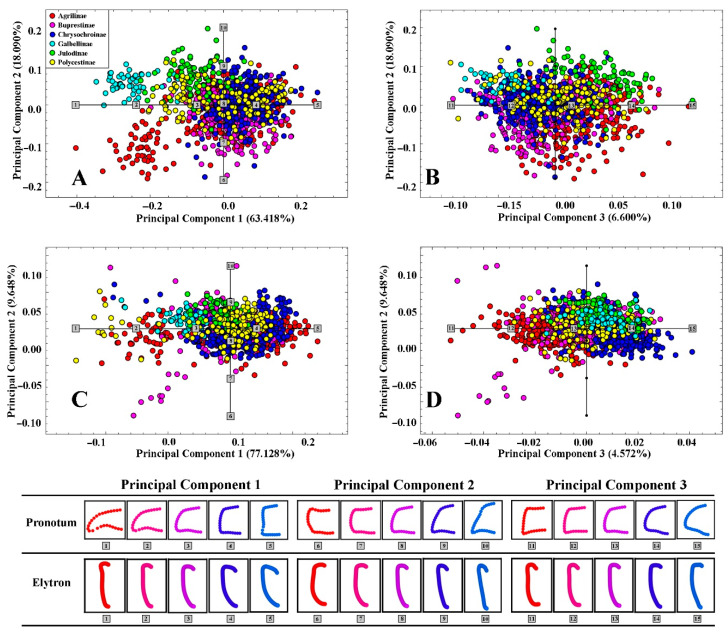
Distribution of subfamilies based on the morphological variation of test traits along principal component axes. Overall, 88.109%/91.348% of observed shape variation was represented by the first three principal components (PCs) in the pronotum and elytron tests, respectively. (**A**): Deformation trend of the pronotum along PC1 and PC2. (**B**): Variation of the pronotum along PC2 and PC3. (**C**)**:** Variation of the elytron along PC1 and PC2. (**D**): Variation of the elytron along PC2 and PC3. All the deformations were showed as shape models in the blocks under the PC-space pictures. Each shape model of variation was calculated at equally spaced intervals along PC axes with coordinates for individual shape model calculations indicated by marks and numbers in the two-dimensional PC shape spaces.

**Figure 4 insects-12-00024-f004:**
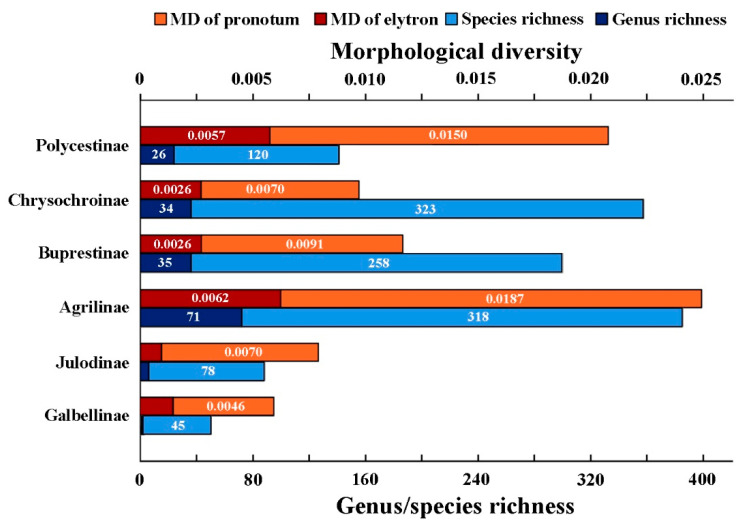
The morphological diversity and species/genus richness among subfamilies. The orange and crimson column represent the morphological diversity of the elytron and pronotum, the “MD” = morphological diversity; the bright and dark blue column represent the genus richness and species richness of each test subfamily, respectively.

**Figure 5 insects-12-00024-f005:**
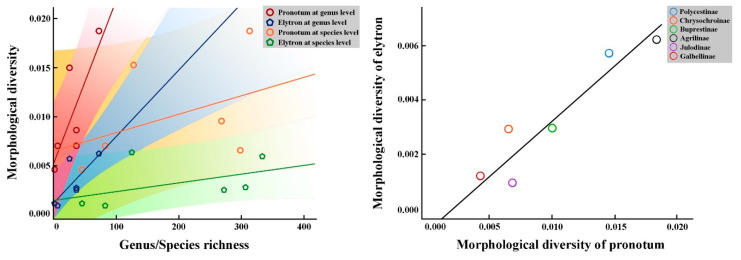
Results of correlation analyses of test subfamilies of Buprestidae. (**A**). Relationships between morphological diversity and genus/species richness. Four sets of measurements were obtained: pronotum test at genus level; pronotum test at species level; elytron test at genus level; elytron test at species level. (**B**). Correlation between the morphological diversity of the pronotum and elytron.

**Table 1 insects-12-00024-t001:** Sampling/described information of test subfamilies.

Test Groups	Sampling Number of Genus	Sampling Number of Species	Described Number of Genus	Described Number of Species
Polycestinae	26	120	82	1255
Chrysochroinae	34	323	112	2742
Buprestinae	35	258	114	3306
Agrilinae	71	318	203	7196
Julodinae	6	78	6	141
Galbellinae	1	45	1	84
Total	173	1106	518	14,724

**Table 2 insects-12-00024-t002:** Morphological diversity (MD) of test traits of subfamilies.

Test Groups	MD of Pronotum	MD of Elytron
Polycestinae	0.0150	0.0057
Chrysochroinae	0.0070	0.0026
Buprestinae	0.0091	0.0026
Agrilinae	0.0187	0.0062
Julodinae	0.0070	0.0009
Galbellinae	0.0046	0.0011

**Table 3 insects-12-00024-t003:** Spearman correlation coefficient between test parameters.

	Genus Richness (*P*)	Species Richness (*P*)
MD of Pronotum	0.714	0.371
MD of Elytron	0.771	0.543

## Data Availability

Data is contained within this article and Appendix A.

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
