# Peer review of "The Relationship between Genus/Species Richness and Morphological Diversity among Subfamilies of Jewel Beetles"

_insects, 2021, doi:10.3390/insects12010024_

Round 1

Reviewer 1 Report

     Authors investigate the relationship of morphological diversity and taxonomic richness at the species and generic levels on the example of speciose and polymorphic family Buprestidae of the world fauna, which is the first research for such a voluminous data. Authors statistically demonstrated a positive correlation between morphological diversity and taxonomic richness, which is an undoubted achievement.

     Unfortunately, it seems that none of the authors is a specialist in the Buprestidae, which led to a number of factual, nomenclatural and taxonomic errors in the main text and supplementary materials (see attached files), which could have been avoided in consultation with a specialist. Necessary information can be also found in “World Catalog and bibliography of the Jewel beetles (Coleoptera: Buprestoidea)”, vol. 1-5 (Bellamy, 2008, 2009) or C.L. Bellamy website: https://cerambycids.com/buprestidae/WorldCat/Classif/webcat.html

Questions: 

  • Line 193: the lowest morphological diversity (0.0010/0.0003) [for Galbellinae]. - Is it possible to evaluate the morphological diversity based on a single species?
  • Lines 375-377: unconformity between species richness and the change trend of morphological diversity in certain groups (e.g., Polycestinae and Chrysochroinae), which was suggested to be caused by the influence of other categories, such as tribe. - How a category can influence the morphological diversity? Inconformity can be resulted from the inadequate sampling within subfamilies (see comments to the Lines 196-197, 202-203).

Comments: 

  • Lines 196-197, 202-203: Its [Polycestinae] elytron diversity (0.0024) was also relatively high as it is in the test of Buprestinae. - Elytron diversity in Polycestinae should be much higher and comparable with Agrilinae due to much more variable body shape (from nearly agriline and buprestine to trachyine in Paratrachys, missing in this study); same is with body size (from ~2 mm in some Haplostethini to 20-30 mm in Polycestini). Perhaps, the results obtained are based on inadequate sampling.
  • Lines 259-260: Usually adult and larval buprestids feed on foliage of the host plant… - It is true only for leaf-miners. Buprestid larvae are endophytes, a great majority are xylophages; Julodinae are exophages and soil habitants. It is quite frequently in Buprestidae that adults and larvae feed on different hosts.
  • Lines 261-262: Xyroscelis crocata (Polycestinae: Xyroscelidini) [61] in Australia which is reported to feed on the sap of the host plant - But the larvae feed within the frond-stalks.
  • Lines 276-277: At the same time, the larvae of this subfamily [Agrilinae] comprise exclusively endophyte species, - Only Julodinae comprise exophyte larvae, all other larval Buprestidae are endophyte.
  • Line 280: with leaf-miner larvae in the Aphanisticini. - Larvae of Aphanisticini are stem-miners.
  • Line 284: more than half of the species [Julodinae] are scattered distributed in a single Regional country (e.g., Julodis algirica, - That reflects the level of our knowledge  in the first place. There are widely distributed species in these genera as well. In any speciose genera a good deal of described species are known from the single country.
  • Line 289: Samples of Galbellinae showed high morphological diversity of the pronotum. - But only one species was studied, how is it possible to conclude about high morphological diversity within the subfamily? And this contradicts the statement about lowest morphological diversity in line 193.
  • Line 291: Although there are few species [Polycestinae] in the  world - There are more than 1255 species in Polycestinae, it’s 4th speciose subfamily of Buprestidae.
  • Line 292: [Polycestinae] are widely distributed in the seven major geographic regions / Lines 300-301: most genera [Polycestinae] are recorded in the six geographic regions except the genus  Blepharum, which is also recorded in Pacific Region - There is also Helferella from Haplostethini in the Pacific region, so seven is correct.
  • Lines 290-295, 299-303. Polycestinae are twice discussed in the different places of the same paragraph with a discussion on Buprestinae between them (lines 295-298). In the same time, Chrysochroinae are completely missed in the Discussion.
  • Lines 302-303: were less than Buprestinae, we found a higher pronotum and elytron morphological diversity in Polycestinae. - But according to Table 2, the elytron MD in Polycestinae (0.0024) is less than in Buprestinae (0.0025).

Editing:

  • Line 19: “jewel species” – should be “jewel beetle species” or only “species” because full name is used in the next line (20).
  • Line 86: Agrilinae – 2 times, should be Agrilinae, Julodinae. In systematic order: 1) Julodinae, 2) Polycestinae, 3)Galbellinae, 4)Chrysochroinae, 5)Buprestinae, 6)Agrilinae (Bellamy, 2003, 2008).
  • Line 94:”comparing the genus/species richness to the total genus/species” - ”comparing the ? tested genus/species richness to the total genus/species”.
  • Line 273: Sumbus – correct for Sambus.
  • Line 298: Melaophila  – correct for Melanophila.

Author Response

Reviewer 1

Authors investigate the relationship of morphological diversity and taxonomic richness at the species and generic levels on the example of speciose and polymorphic family Buprestidae of the world fauna, which is the first research for such a voluminous data. Authors statistically demonstrated a positive correlation between morphological diversity and taxonomic richness, which is an undoubted achievement.

Unfortunately, it seems that none of the authors is a specialist in the Buprestidae, which led to a number of factual, nomenclatural and taxonomic errors in the main text and supplementary materials (see attached files), which could have been avoided in consultation with a specialist. Necessary information can be also found in “World Catalog and bibliography of the Jewel beetles (Coleoptera: Buprestoidea)”, vol. 1-5 (Bellamy, 2008, 2009) or C.L. Bellamy website: https://cerambycids.com/buprestidae/WorldCat/Classif/webcat.html

Questions: 

  1. Line 193: the lowest morphological diversity (0.0010/0.0003) [for Galbellinae]. - Is it possible to evaluate the morphological diversity based on a single species?

Author’s response: We agree with the reviewer. We have discussed the topic extensively with Dr. Haitian SONG, a specialist on Buprestidae, then we believe it is necessary to increase the test sample of Galbellinae. We have increased the test samples of this group from 1 to 45 (about 84 species are known around the world) with the help of buprestid experts and related documents, which largely reflects the morphological information of this group (see Line 122-124 of the major revision-manuscript, also see the Table 1, please). Through the analysis of geometric morphology, we found that the morphological diversity of the elytra changes with the increase of sample size more than that of pronotum (see Table 2, please). At the same time, we also performed geometric morphological and statistical analysis of the additional data, and updated all the illustrations and tables in the major revision-manuscript.

  1. Lines 375-377: unconformity between species richness and the change trend of morphological diversity in certain groups (e.g., Polycestinae and Chrysochroinae), which was suggested to be caused by the influence of other categories, such as tribe. - How a category can influence the morphological diversity? Inconformity can be resulted from the inadequate sampling within subfamilies (see comments to the Lines 196-197, 202-203).

Author’s response: Thanks for the suggestions. The original intention of this sentence is to express that the taxonomic category will affect the test results of biological diversity (in this article, the correlation between MD and SR/GR is used as a display) instead of affecting the morphological diversity, and we will consider the tribe-level test in follow-up research. We have re-written this part (see Line 557-557 of the major revisions, please).

Comments: 

  1. Lines 196-197, 202-203: Its [Polycestinae] elytron diversity (0.0024) was also relatively high as it is in the test of Buprestinae. - Elytron diversity in Polycestinae should be much higher and comparable with Agrilinae due to much more variable body shape (from nearly agriline and buprestine to trachyine in Paratrachys, missing in this study); same is with body size (from ~2 mm in some Haplostethini to 20-30 mm in Polycestini). Perhaps, the results obtained are based on inadequate sampling.

Author’s response: We agree with the reviewers and thank them for this suggestion. We have discussed the topic extensively with Prof. Aimin Shi, a specialist on Buprestidae, and subsequently we added 10 species of Paratrachys (about 25 species have been described from all over the world) into the test dataset and also analyzed the morphological diversity of added dataset of Polycestinae (see Line 122-124 of the major revisions, also see the Table 1, please). Consistent with the reviewer's point of view, the addition of morphological information of Paratrachys greatly affected the results of the elytron morphological diversity of Polycestinae, which also makes our research more robust and reliable. Furthermore, the analysis method of geometric morphometrics can simply obtain shape information by eliminating the factors such as the size and direction of the test sample, but we also appreciate the suggestion and we believe that adding the body-length information of the sample in the follow-up research will provide more interesting findings.

  1. Lines 259-260: Usually adult and larvalbuprestids feed on foliage of the host plant… - It is true only for leaf-miners. Buprestid larvae are endophytes, a great majority are xylophages; Julodinae are exophages and soil habitants. It is quite frequently in Buprestidae that adults and larvae feed on different hosts.

Author’s response: Thanks for the suggestions. We have discussed the topic extensively with Prof. Aimin Shi, a specialist on Buprestidae and re-written this part (see Line 383, please).

  1. Lines 261-262: Xyroscelis crocata (Polycestinae: Xyroscelidini) [61] in Australia which is reported to feed on the sap of the host plant - But the larvae feed within the frond-stalks.

Author’s response: We agree with the reviewer. We have discussed the topic extensively with Prof. Aimin Shi and Dr. Haitian SONG, specialists on Buprestidae to ensure that the information about the larvae habits of Xyroscelis are correct, and re-written this part (see Line 386-387, please). Regarding the feeding habits of adults, we obtained information from Bellamy (1997) that this group feeds on plant sap of Macrozamia communis.

  1. Lines 276-277: At the same time, the larvae of this subfamily [Agrilinae] comprise exclusively endophyte species, - Only Julodinae comprise exophyte larvae, all other larval Buprestidae are endophyte.

Author’s response: Thanks for the suggestions. We originally wanted to express the special habit of the buprestid larvae (the endophyte). Sorry for the confusion caused to the reviewer. We have corrected this error (see Line 401, please).

  1. Line 280: with leaf-miner larvae in the Aphanisticini. - Larvae of Aphanisticini are stem-miners.

Author’s response: We agree with the reviewer. We have discussed the topic extensively with Prof. Aimin Shi and Dr. Haitian SONG, two specialists on Buprestidae to ensure that information about the larvae habits are correct. We have corrected this error (see Line 404, please).

  1. Line 284: more than half of the species [Julodinae] are scattered distributed in a single Regional country (e.g., Julodis algirica, - That reflects the level of our knowledge in the first place. There are widely distributed species in these genera as well. In any speciose genera a good deal of described species are known from the single country.

Author’s response: Thanks for the suggestions. After discussing the topic with Dr. Aimin Shi, a specialist on Buprestidae and checking the related articles from Bellamy, we find that this description of these species is arbitrary and have re-written this part (see Line 408-409, please).

  1. Line 289: Samples of Galbellinae showed high morphological diversity of the pronotum. - But only one species was studied, how is it possible to conclude about high morphological diversity within the subfamily? And this contradicts the statement about lowest morphological diversity in line 193.

Author’s response: We agree with the reviewer. We have discussed the topic extensively with Prof. Aimin Shi, a specialist on Buprestidae. Following this, we believed it to be necessary to increase the test sample of Galbellinae. The test samples of this group in this study have increased from 1 to 45 with the help of buprestid experts and related documents, which largely reflects the morphological information of this group (see Line 122-124 of the major revision-manuscript, also see the Table 1, please) and also re-written this part (see the Line 252, please).

  1. Line 291: Although there are few species [Polycestinae] in the world - There are more than 1255 species in Polycestinae, it’s 4th speciose subfamily of Buprestidae.

Author’s response: Thanks for the suggestions. We confirmed the improper use of this word here and have rewritten this sentence (see Line 449-455, please).

  1. Line 292: [Polycestinae] are widely distributed in the sevenmajor geographic regions / Lines 300-301: most genera [Polycestinae] are recorded in the six geographic regions except the genus Blepharum, which is also recorded in Pacific Region - There is also Helferella from Haplostethini in the Pacific region, so seven is correct.

Author’s response: We agree with the reviewer. We have re-checked the related information of Polycestinae from ‘World Catalog and bibliography of the Jewel beetles (Coleoptera: Buprestoidea)’ and re-written this part (see Line 449-455, please).

  1. Lines 290-295, 299-303. Polycestinae are twice discussed in the different places of the same paragraph with a discussion on Buprestinae between them (lines 295-298). In the same time, Chrysochroinae are completely missed in the Discussion.

Author’s response: We agree with the reviewer’s comments on repetitive descriptions of this group causing confusion in content. We have re-written this part and also add the discussion about Chrysochroinae (see Line 444-448, please).

  1. Lines 302-303: were less than Buprestinae, we found a higher pronotum and elytron morphological diversity in Polycestinae. - But according to Table 2, the elytron MD in Polycestinae (0.0024) is less than in Buprestinae (0.0025).

Author’s response: We agree with the reviewer. After we added test samples of the genus Paratrachys according to the reviewer’s suggestions, we found the uniqueness of this genus and its important role in the morphological diversity of elytron in Polycestinae (see Table 2, please). Through the geometric morphometrics, we obtained a higher elytron morphological diversity in Polycestinae (0.0057) compared with it in Buprestinae (0.0025), so this sentence is valid in the current manuscript.

Editing:

  1. Line 19: “jewel species” – should be “jewel beetle species” or only “species” because full name is used in the next line (20).

Author’s response: We agree with the reviewer, and the word ‘jewel’ has been deleted following the suggestion (see Line 35, please).

  1. Line 86: Agrilinae – 2 times, should be Agrilinae, Julodinae. In systematic order: 1) Julodinae, 2) Polycestinae, 3) Galbellinae, 4) Chrysochroinae, 5)Buprestinae, 6)Agrilinae (Bellamy, 2003, 2008).

Author’s response: Thanks for the suggestion. This sentence has been re-written (see Line 124, please).

  1. Line 94: “comparing the genus/species richness to the total genus/species” - ” comparing the ? testedgenus/species richness to the total genus/species”.

Author’s response: Thanks for the suggestion. Sorry that our wording has caused confusion to the reviewer, we have re-written this part (see Line 142, please).

  1. Line 273: Sumbus– correct for 

Author’s response: We agree with the reviewer, and have changed the word (see Line 398, please).

  1. Line 298: Melaophila – correct for Melanophila.

Author’s response: We agree with the reviewer, and have changed the wording (see Line 417, please).

Reviewer 2 Report

Compared to the previous version I reviewed, the manuscript was significantly re-worked, largely in the directions suggested in my previous review. I really appreciate the effort of the authors to improve the study, because I really liked the topic and the idea which in my opinion is very interesting and original. Also, I can notice that a native-speaking researcher was included as a co-author, and can confirm that the level of English and the clearness of the writing improved largely. 

The study as it is now, is nearly ready to be published, and below I am providing just few short comments which came to my mind while reading the current version:

  • What is the y axis in Fig. 1. It says "proportions of groups" which however means not much. From the text I understand that it is proportion of studied versus described taxa? And is the 100% (i.e. number of described taxa) taken as the total for Buprestidae, or a total for each group. From the values I would guess the total of Buprestidae, but I guess this is not what you wanted to show? I think the information you wanted to show is what part of the diversity of each group did you sample (as this has consequences for your conclusions: it makes difference if you sampled a single Galbelinae which however is the only genus and one of few Galbelinae species, as in that case even a single species covers lots (or all) morphological diversity of the group, or you sampled many Agrilinae which however cover 50% of genera and 20% of species, which indicates that some additional morphological diversity may be still present). But maybe I am wrong, in any case adapt the legend to y-axis to reflect what you measured more adequately.
  • In results you write that along one PCA axis the elytron became larger (e.g. row 166). Please adapt that, as "larger" reflects to body size, which is not considered in your analyses (it was actually purposely filtered out so you can compare the shape of tiny species to large ones). What you are comparing are proportions, not sizes!
  • I am not a statistician, but does it make sense to make correlation tests with p=0? Goes little bit against my understanding of how p is used - being very low means that you can be only wrong with a very low probability, and p<0.05 or p<0.005 are usually used as treshholds reported (i.e. if your p is close to 0, it would fall into p<0.005 treshhold)? But as I say, its just a feeling, I am not strong in statistics.
  • In discussion (line 244) you write that the pronotum shape - species diversity is negatively correlated... how that can be, when you correlation coeficients are all positive, and all your lines in Fig. 5 grow from left to right (i.e. with higher taxon diversity, you always have higher morphological diversity). Negative corelation would mean that higher taxon diversity means lower morphological diversity, which is not what you observe!
  • Interpretation - you write that your results are hard to interpret. But I think some conclusions can be done: (1) pronotum shape diversity correlates strongly with genera, not with species = pronotal shape is more conserved between species, and only varies among less related clades. Consequently, I would expect that part of generic definitions are hence based on pronotal shape, which would explain the nice correlation you found. (2) In contrast, elytra seem to be less conserved in morphology, varying largely also between species within the genera. Seems that pronotum is more "conservative" in shape than elytra, possibly under stronger stabilizing selection - hard to say why, maybe it can be due to life style or some developmental constraints). In contrast, elytra seem to be more free of evolutionary constraints and hence can vary more freely even among species.
  • In this context, it would be very interesting to see whether the morphological variability correlate to ecological variability in each group. You mention several ecological guilds in Discussion (like xylophagy - leaf miners - stem miners) - are the subfamilies with larger morphological diversity (Polycestinae, Agrilinae) more diverse in that than subfamilies which are diverse in species but limited in morphological diversity (e.g. Chrysochloinae, Buprestinae)?

I am looking forward to see this manuscript published!

Author Response

Reviewer 2

Compared to the previous version I reviewed, the manuscript was significantly re-worked, largely in the directions suggested in my previous review. I really appreciate the effort of the authors to improve the study, because I really liked the topic and the idea which in my opinion is very interesting and original. Also, I can notice that a native-speaking researcher was included as a co-author, and can confirm that the level of English and the clearness of the writing improved largely.

The study as it is now, is nearly ready to be published, and below I am providing just few short comments which came to my mind while reading the current version:

  1. What is the y axis in Fig. 1 It says "proportions of groups" which however means not much. From the text I understand that it is proportion of studied versus described taxa? And is the 100% (i.e. number of described taxa) taken as the total for Buprestidae, or a total for each group. From the values I would guess the total of Buprestidae, but I guess this is not what you wanted to show? I think the information you wanted to show is what part of the diversity of each group did you sample (as this has consequences for your conclusions: it makes difference if you sampled a single Galbelinae which however is the only genus and one of few Galbelinae species, as in that case even a single species covers lots (or all) morphological diversity of the group, or you sampled many Agrilinae which however cover 50% of genera and 20% of species, which indicates that some additional morphological diversity may be still present). But maybe I am wrong, in any case adapt the legend to y-axis to reflect what you measured more adequately.

Author’s response: Thanks for the suggestions. About the ‘proportions of groups in Fig. 1’: since we have shown the sampling numbers of all taxa in Table 1, Figure 1 is mainly to provide a more intuitive display of the sampling rate in the test/described dataset. Figure 1 shows the proportion of each taxa in all taxa in the test/described dataset. For example, regarding the sampling rate test of Polycestinae, we first calculate the proportion of the genera/species of this subfamily in all the test genera/species (the number of genera/species of Polycestinae versus the number of genera/species of the test Buprestidae), then compare the genera/species sampling ratio with it in the described dataset for indicating the objectivity of our experimental methods. Furthermore, for the part of the sampling size of Galbelinae, we have discussed the topic extensively with Prof. Aimin Shi (a specialist on Buprestidae), after which we believe it is necessary to increase the test sample of Galbellinae to ensure that the test results of this subfamily are sufficiently objective. The test samples of this group in this study have increased from 1 to 45 (84 species known around the world) with the help of buprestid experts and related documents, which largely reflects the morphological information of this group (see Line 123-125 of the major revision-manuscript, also see the Table 1, please). Through a new round of analysis, we found that the increase in test samples of Galbelinae changed the morphological diversity of the pronotum and elytron, but it did not change our original conclusion: the morphological diversity of the pronotum is higher than it of elytron; higher taxa perform better in the correlation between morphological diversity and genus/species richness. At the same time, we also performed geometric morphological and statistical analysis of the added data, and updated all the illustrations and tables in the major revision-manuscript.

  1. In results you write that along one PCA axis the elytron became larger (e.g. row 166). Please adapt that, as "larger" reflects to body size, which is not considered in your analyses (it was actually purposely filtered out so you can compare the shape of tiny species to large ones). What you are comparing are proportions, not sizes!

Author’s response: We agree with the reviewer. And we have modified the word ‘larger’ to ‘more prominent’ (see Line 221-222, please).

  1. I am not a statistician, but does it make sense to make correlation tests with p=0? Goes little bit against my understanding of how p is used - being very low means that you can be only wrong with a very low probability, and p<0.05 or p<0.005 are usually used as treshholds reported (i.e. if your p is close to 0, it would fall into p<0.005 treshhold)? But as I say, its just a feeling, I am not strong in statistics.

Author’s response: Thanks for the suggestions. We listed the concrete p-value in Table 3 to show the significance level of all test groups originally. After discussion, we agree with the reviewer and believe that this kind of display is of little significance in expressing the correlation between test parameters, so we refer the expression of the relevant p-value to a comparison with the threshold at the 0.05 level. At the same time, with the addition of the test data of this manuscript , we have obtained the p-value in the correlation between elytron MD and GR that is not zero (see Line 294-296, also see the Table 3, please).

  1. In discussion (line 244) you write that the pronotum shape - species diversity is negatively correlated... how that can be, when you correlation coeficients are all positive, and all your lines in Fig. 5 grow from left to right (i.e. with higher taxon diversity, you always have higher morphological diversity). Negative corelation would mean that higher taxon diversity means lower morphological diversity, which is not what you observe!

Author’s response: Thanks for the suggestions. The word ‘negatively’ here is a writing error. Based on the test taxa and data of this research (according to the suggestion of the buprestid experts, the relevant test samples of the Galbelinae and Paratrachys in Polycestinae were added), we found that there was no statistical significance (p>0.05) (not negative correlation) in the species-level test based on Spearman correlation coefficient analysis (see Line 296, please). At the same time, we visualize the correlation between MD and GR/SR under different conditions (different test traits and categories) by showing the distribution of test groups Figure 5. We find that the distribution of test groups in the genus-level test is relatively concentrated, while the distribution of test groups in the species-level test is relatively random, which shows that there is no correlation between the test parameters. We have re-written this part and re-work the illustration to make it clearer (see Line 304-305, please).

Interpretation

  1. you write that your results are hard to interpret. But I think some conclusions can be done: (1) pronotum shape diversity correlates strongly with genera, not with species = pronotal shape is more conserved between species, and only varies among less related clades. Consequently, I would expect that part of generic definitions are hence based on pronotal shape, which would explain the nice correlation you found. (2) In contrast, elytra seem to be less conserved in morphology, varying largely also between species within the genera. Seems that pronotum is more "conservative" in shape than elytra, possibly under stronger stabilizing selection - hard to say why, maybe it can be due to life style or some developmental constraints). In contrast, elytra seem to be more free of evolutionary constraints and hence can vary more freely even among species.

Author’s response: Thanks for the suggestions. We have added 10 species of Paratrachys (about 25 species have been described over the world) which has significant morphological variances from the discussion with other buprestid specialists and 44 species of Galbellinae (84 species known around the world) into the test dataset, and also analyzed the morphological diversity of the updated dataset (see Line 123-125 and also see the Table 1, please), then we confirmed our previous conclusion: the MD of pronotum is higher than it in elytron and the correlation between MD of elytron and GR was higher than it between MD of pronotum and GR. We agree with the points of reviewer, based on the current test dataset and the results, the pronotum evolved under stronger under stronger stabilizing selection. However, due to the limitation of test features and test categories, we cannot fully confirm the influencing factors of this phenomenon. We hope the following test with more sufficient features and more comprehensive groups may gradually explain it.

  1. In this context, it would be very interesting to see whether the morphological variability correlate to ecological variability in each group. You mention several ecological guilds in Discussion (like xylophagy - leaf miners - stem miners) - are the subfamilies with larger morphological diversity (Polycestinae, Agrilinae) more diverse in that than subfamilies which are diverse in species but limited in morphological diversity (e.g. Chrysochloinae, Buprestinae)?

Author’s response: Thanks for the suggestions. We have re-written the ecological part and try to show the correlation between our test results and the buprestid ecological variability after we added the dataset of Paratrachys and Galbellinae. We also found that the ecological guilds of some taxa with larger MD more diverse than it in the taxa with limited MD, for example, the Agrilinae has three habits: xylophagous type, stem-miners and leaf-miners, while the Julodinae comprise exophyte larvae. Our current higher-taxa biodiversity research and the particularity of buprestid ecology in different groups make it difficult to clearly compare these habits of all taxa. In follow-up research, we will conduct targeted group research and take biological factors into consideration.

Round 2

Reviewer 1 Report

Authors substantially improved the previous version of the MS and added taxa proposed by the reviewer, so that only a few easily fixable errors were revealed in the MS main text. Unfortunately, the authors did not comply with the previous recommendation to verify not only the main text, but also the complementary file (the list of studied Buprestid taxa) for the validity and spelling of taxa names and their current systematic position (according to Bellamy, 2008, 2008 or his web-site https://cerambycids.com/buprestidae). For example, among Polycestinae, some of the species of Polyctesis actually belong to the genus Schoutedeniastes, and most species of Tyndaris belong to the genus Paratyndaris; among Chrysochroinae, most species of Psiloptera belong to the genus Lampetis, the genus Chrysobothris (C. affinis C. solieri, lines 166-167, previous version) and Diadoxus belong to Buprestinae; among Buprestinae, all mentioned species of Ovalisa and Poecilonota (except P. variolosa) currently placed in the genus Lamprodila, which belongs to Chrysochroinae; etc. The list also includes a large number of synonyms and subspecies. Although these errors are unlikely to greatly distort the results obtained, I would advise to make necessary corrections to the supplementary file for the final version.

Main text.

Line 4 [key words]: [jewel] beetle – [jewel] beetles.

Lines 205-206: The Agrilinae exhibited…………..with the highest genus richness and fairly high species richness (Table 2,… – Agrilinae exhibited highest genus and species richness (Agrilus is a most speciose genus in the Animal kingdom [38]).

Lines 209-210: [Galbellinae] was found to have the lowest pronotum morphological diversity (0.0046) – in lines 309-310 it is proclaimed that "Samples of Galbellinae showed high morphological diversity of the pronotum". What is correct?

Line 212:  high pronotum diversity and elytron (0.0150/0.0057) –high  pronotum and elytron diversity...

Lines 213-214: Its [Polycestinae] elytron diversity (0.0024) was also relatively high as it is in the test of Buprestinae. – But in the replaced table it is 0.0057 which is much higher than in Buprestinae (0.0025).

Lines 328-329: In particular, the posterior angle of the pronotum and the scutellum edge attached to the anterior ledge of elytron, not only in the horizontal or vertical extension. – In what extension else? What do you mean?

Lines 382-384, 395, 431: At the same time, taking Buprestidae as an example, we found that there was a strong correlation between the morphological diversity of subfamilies and genus richness, … – But Polycestinae demonstrate higher MD and lower GR comparing to Chrysochroinae and Buprestinae, thus the correlation is not so strong.

Line 534: 35. Kukalovapeck, J.; Lawrence, – Kukalova-Peck, J.;

Lines 536, 610:  36 [67]. Evans, A.M.; – same reference twice.

Author Response

Reviewer 1

Authors substantially improved the previous version of the MS and added taxa proposed by the reviewer, so that only a few easily fixable errors were revealed in the MS main text. Unfortunately, the authors did not comply with the previous recommendation to verify not only the main text, but also the complementary file (the list of studied Buprestid taxa) for the validity and spelling of taxa names and their current systematic position (according to Bellamy, 2008, 2008 or his web-site https://cerambycids.com/buprestidae). For example, among Polycestinae, some of the species of Polyctesis actually belong to the genus Schoutedeniastes, and most species of Tyndaris belong to the genus Paratyndaris; among Chrysochroinae, most species of Psiloptera belong to the genus Lampetis, the genus Chrysobothris (C. affinis C. solieri, lines 166-167, previous version) and Diadoxus belong to Buprestinae; among Buprestinae, all mentioned species of Ovalisa and Poecilonota (except P. variolosa) currently placed in the genus Lamprodila, which belongs to Chrysochroinae; etc. The list also includes a large number of synonyms and subspecies. Although these errors are unlikely to greatly distort the results obtained, I would advise to make necessary corrections to the supplementary file for the final version.

Author’s response: Thanks for the suggestions. In the last version we confused the suggestions on the taxonomy-status check of the test samples (in the supplement) with it in the main text and didn’t check it in supplement, very sorry for the confusion caused to the reviewer.

In this round of the revision, we strictly checked all the names of species (1215 test beetles) in the supplement based on the literatures and jewel beetles’ website of Bellamy, and revised all the names of subspecies and the taxonomic status of genera. At the same time, we also checked and corrected the wrong attribution of those special groups the Reviewer 1 mentioned in the comments.

After checking the taxonomic status and names of all the test samples, we conducted a relevant analysis of the new dataset. As the Reviewer 1 said, we found that the changes in some taxa did not affect our original research conclusions. In addition, we readjusted the figures and tables in this manuscript (all tables and figures except Figure 2 which shows the test curves on pronotum and elytron) based on the new dataset analysis, and rechecked the content of the manuscript again to avoid recurring errors in the data description.

Questions: 

  1. Line 4 [key words]: [jewel] beetle – [jewel] beetles.

Author’s response: We agree with the reviewer, and have changed the word (see Line 53, please).

  1. Lines 205-206: The Agrilinae exhibited…………..with the highest genus richness and fairly high species richness (Table 2,… – Agrilinae exhibited highest genus and species richness (Agrilus is a most speciose genus in the Animal kingdom [38]).

Author’s response: Thanks for the suggestions. We originally wanted to express the test species richness not the status of described species richness around the world in Agrilinae. Sorry for the confusion caused to the reviewer, we believe this confusion is affected by the sampling method of this study (the sampling rate of the test groups is fairly reasonable on the whole, through the Figure 1). We have corrected this error (see Line 253, please).

  1. Lines 209-210: [Galbellinae] was found to have the lowest pronotum morphological diversity (0.0046) – in lines 309-310 it is proclaimed that "Samples of Galbellinae showed high morphological diversity of the pronotum". What is correct?

Author’s response: Thanks for the suggestions. We have re-checked the relevant content and confirmed the writing error of this vocabulary: ‘high’ in Line 309. The original intention here is to say that Galbellinae has only one genus with 84 species, most of which are concentrated in the Afrotropical region, resulting in low morphological diversity of this group. We have corrected this error (see Line 407-408, please).

  1. Line 212:  high pronotum diversity and elytron (0.0150/0.0057) –high  pronotum and elytron diversity...

Author’s response: We agree with the reviewer, and have re-written this sentence (see Line 275, please).

  1. Lines 213-214: Its [Polycestinae] elytron diversity (0.0024) was also relatively high as it is in the test of Buprestinae. – But in the replaced table it is 0.0057 which is much higher than in Buprestinae (0.0025).

Author’s response: We agree with the reviewer. We have re-checked the relevant content and table, and confirmed the writing error of the number ‘0.0024’ which was not modified after adding the test sample size in the previous version of manuscript. Sorry for the confusion caused to the reviewer, we have corrected this error and re-written this sentence (see Line 276, please).

  1. Lines 328-329: In particular, the posterior angle of the pronotum and the scutellum edge attached to the anterior ledge of elytron, not only in the horizontal or vertical extension. – In what extension else? What do you mean?

Author’s response: Thanks for the suggestions. We originally wanted to express that the results got through geometric morphometrics reflected the relevance of these traits (pronotum, scutellum and elytron) in terms of the morphology and function, rather than just showing the size or morphological changes of a single trait by the traditional testing method. We have re-written this part (see Line 435-436, please).

  1. Lines 382-384, 395, 431: At the same time, taking Buprestidae as an example, we found that there was a strong correlation between the morphological diversity of subfamilies and genus richness, … – But Polycestinae demonstrate higher MD and lower GR comparing to Chrysochroinae and Buprestinae, thus the correlation is not so strong.

Author’s response: We agree with the reviewer, we also find this word doesn’t warrant here and have re-written this part (see Line 491, Line 504 and Line 542, please).

  1. Line 534: 35. Kukalovapeck, J.; Lawrence, – Kukalova-Peck, J.;

Author’s response: Thanks for the suggestions, and we have corrected this writing error (see Line 647, please).

  1. Lines 536, 610:  36 [67]. Evans, A.M.; – same reference twice.

Author’s response: We agree with the reviewer, and have corrected this writing error (see Line 649 and Line 723, please).

This manuscript is a resubmission of an earlier submission. The following is a list of the peer review reports and author responses from that submission.

Round 1

Reviewer 1 Report

I am not an expert in the geometric morphometrics and statistcs (like none of the authors seems are the specialists in the systematics of this particular group), so my comments concern mainly the Buprestid taxonomy and distribution as well.

Line 146. "genus richness and species richness was 82 and 6 respectively).

- Maybe vice versa: 6 and 82 82 respectively.

Line 240. "...[Galbellinae] this group was relatively conservative...".

-But Galbellinae is one of the most morphologically advanced groups of Buprestidae, particularly concerning their prothoracic structure. [56] is a book on American Beetles whereas Galbellinae lacking in America.

Line 244-245. "... [Julodinae] many species are distributed from the Southernn Palaearctic and Oriental regions to the Cape region of South Africa."

- There are no common species for Oriental / Palaearctic regions and Afrotropics, only few species occurring close to the regional border between South Palaerctic and Afrotropics are recorded in both regions (mainly in North Africa and Sahel).

Line 246. "The largest two genera, Julodis and Sternocera, are recorded from the west part of Pakistan to Southeast Asia [56]."

-  Julodis species are lacking in Southeast Asia. Only few species reaching western (desert) parts of Pakistan and India. Only one species of Sternocera reaches the eastern limits of Iran and Pakistan in Asia. Julodinae do not occur in America, so reference to the book on American beetles [56] is not a good deal.

Line 256. "Since the pronotum is not restricted by flight machinery..."

Actually,  posterior margin of pronotum is adapted to fix the elytral base in repose and in flight and develops a rather complicated locking mechanism.

Reviewer 2 Report

 This manuscript was well-prepared. The correlation between the category richness and morphological diversity is interesting and novel. ANd there are some aspects to be discussed:

  1. Is there some effect by sexual dimorphism of buprestidae?
  2. Do the sampling method and representativeness have an impact on the results?
  3. The explanation in the discussion part is not direct, so the logic should be strengthened.

Reviewer 3 Report

The morphological diversity is surely the aspect of immense insect diversity which is best known to the public. Despite of that, it is very rarely studied, and its relationship to taxonomic diversity is an exciting topic! This was also the reason why I agreed to review this manuscript, as its central idea is simple but awesome and really original: to take the data on body shape morphology across the whole family Buprestidae and test its correlation to the taxonomic diversity.

Unfortunately, when I started to read the paper, I got very quickly very disappointed – the central idea is great, but the way in which the study is done is very unlucky. It combines few serious problems:

  • The style of writing is very unorganized, many statements are unclear, many sentences are meaningless. This is combined with rather poor quality of English (sometimes in grammar, but more frequently in improperly chosen words). Many sentences are not even finalized (they miss verbs or important parts of the statements). In many parts, the texts looks like written by a statistician without having an idea that the study is about beetles (e.g., species are often called “samples” for example, the statement “category richness” is totally unclear and in fact goes for number of species or genera, etc.). All this combined makes the text very difficult to read and understand, and a total rewriting will be necessary.
  • Despite the central idea is simple and really cool, they way how the data are analyzed is kind of random, making me feeling like the authors randomly played with the data to get some results, without actually having an idea what is their question to reply. For example, the test of sampling ration is a total non-sense, I do not understand why these were done and what the graph on Fig. 1 should illustrate. Why do you suppose the taxon diversity in each subfamily should be anyhow correlated (i.e. what the lines in the graph actually say?). The proportion of species/genera sampled is surely an important description of the dataset, should be present as a table, but no “analysis” of these data is needed. The subsequent analyses are more meaningful, yet kind of too simple – in fact you quantified the morphological diversity within each subfamily and put in in relationship to the number of species or genera in that subfamily. That is fine, but it should be not the only question asked, and it should not be presented as 4 independent analyses (as it is in fact a single analysis with different input data).
  • The graphical presentation looks nice at the first view, but is full of formal flaws – I provided detailed comments to each picture in the PDF, but as an example I can take the axis and bars in Fig. 2 (morphological diversity and species richness have totally different units and hence cannot have one axis, the morphological diversity needs to be illustrated by bars reflecting the value, i.e. being small in color area for small values and large for larger values, now it is the opposite). In Fig. 3, the shapes explaining the PC axes cannot overlap with the data points, as then the graphs make no sense.
  • The morphometric part needs a better explanation as it concerns the methods, this part is very brief and does not allow to repeat the analysis. For example, pseudolandmarks are always situated on a line between two usual landmarks which need to be homologous… these landmarks are not even mentioned, and their homology is not explained and justified.
  • The discussion in generally does not really provide any explanation of the observed trends. My feeling is that the authors do not actually have any insight into the family Buprestidae and hence failed to discuss the relevance of their results in a meaningful way. Instead, the discussion is a set of random statements without real relationship to the topic studied (e.g. how the distribution of some subfamilies is supposed to affect the morphological diversity?). I strongly recommend to include a specialist for the Buprestidae who knows these beetles well, I am sure he or she will be able to offer some hypotheses on the trends observed, based on the knowledge of biology of the particular group.

For all reasons listed above, I have to request the rejection of the manuscript in its current form. However, I am still excited by the basic idea, and believe that the dataset you have is really powerful in examining the evolution of morphological diversity in jewel beetles. Below I am offering some quick ideas of possible improvements or questions to ask:

  • Call species richness and number of genera by its real names, do not try to cover them by “category richness” which is a term having no sense and actually confusing the reader. Using number of species is fine, since species are basic units of diversity. For higher ranks, like genera and subfamilies, you should keep in mind that these categories are artificial, reflecting more the way the traditional taxonomists treat the diversity of the family rather than the evolution of the group. It is still fine to use these categories, but I would expect some information as it concerns the “reality” of these ranks. Is there some analysis confirming that the six subfamilies are really monophyletic units (in other words, do you work with natural units of diversity or is there a danger that this can be biased by the fact that some traditional subfamilies are in fact polyphyletic?). Same for genera – I understand that the monophyly of most genera is not tested, but are these some indications that some genera may in fact be polyphyletic due to similar morphology or lifestyle? It is also fine to use genera as they are defined now, but the fact that this may be unnatural and hence biased number needs to be discussed and considered.
  • Since most genera are likely defined to group morphologically similar species, it may be expected that most of the morphological diversity is “between genera” rather than “within genera”. Since you sampled 1000+ species of less than 200 genera, I guess you have multiple species per genus in most genera, so you can test this very easily. Are there some large genera which are more diverse morphologically than others? Can this be explained by number of species in the genus, in a similar way as you explain now for the subfamilies? In other words, if you do the graph as in Fig. 4 but put one point for each genus (or selected larger genera for which you have more than few species sampled), will the trend be the same?
  • Is the shape of pronotum (and its diversity within genus or subfamily) somehow constrained or correlated to the shape of elytron (and its diversity within the genus or subfamily)? I would guess that a simple graph of morphological diversity of pronotum (on X axis) and elytra (on Y axis) can show interesting patterns. You should also try it for subfamilies (i.e. 6 points) but also larger genera (i.e. many more points).
  • What are (or can be) the drivers or constraints of body shape (or only elytra shape, or only pronotum shape) in various groups? If you take the biology into account, does it make sense that some genera are more uniform in body shape and some more diverse? Are the genera which are exceptional for their very high or very low morphological diversity somehow exceptional in lifestyle, number of species, age of origin or anything else?

I am sure these are not all possible questions you can easily test with your cool dataset: I am absolutely positive that the discussion with an actual specialist on buprestids may bring many more interesting ideas. And that after the analyses are redone with particular questions in mind or particular hypotheses to test, it will results in a really great and influential publication!
